# Level II Oncoplastic Surgery as an Alternative Option to Mastectomy with Immediate Breast Reconstruction in the Neoadjuvant Setting: A Multidisciplinary Single Center Experience

**DOI:** 10.3390/cancers14051275

**Published:** 2022-03-01

**Authors:** Alba Di Leone, Antonio Franco, Daniela Andreina Terribile, Stefano Magno, Alessandra Fabi, Alejandro Martin Sanchez, Sabatino D’Archi, Lorenzo Scardina, Maria Natale, Elena Jane Mason, Federica Murando, Fabio Marazzi, Armando Orlandi, Ida Paris, Giuseppe Visconti, Antonella Palazzo, Valeria Masiello, Liliana Barone Adesi, Marzia Salgarello, Riccardo Masetti, Gianluca Franceschini

**Affiliations:** 1Breast Unit, Department of Women, Children and Public Health Sciences, “A. Gemelli” IRCCS, 00168 Roma, Italy; alba.dileone@policlinicogemelli.it (A.D.L.); danielaandreina.terribile@policlinicogemelli.it (D.A.T.); stefano.magno@policlinicogemelli.it (S.M.); martin.sanchez@policlinicogemelli.it (A.M.S.); sabatino.darchi@policlinicogemelli.it (S.D.); lorenzoscardina@libero.it (L.S.); maria.natale@policlinicogemelli.it (M.N.); elenajanemason@gmail.com (E.J.M.); murandofederica@gmail.com (F.M.); riccardo.masetti@policlinicogemelli.it (R.M.); gianluca.franceschini@policlinicogemelli.it (G.F.); 2Precision Medicine Breast Unit, Scientific Directorate, Department of Women, Children and Public Health Sciences, “A. Gemelli” IRCCS, 00168 Roma, Italy; alessandra.fabi@policlinicogemelli.it; 3Cancer Radiation Therapy, Department of Diagnostic Imaging, Oncological Radiotherapy and Hematology, “A. Gemelli” IRCCS, 00168 Roma, Italy; fabio.marazzi@policlinicogemelli.it (F.M.); valeria.masiello@policlinicogemelli.it (V.M.); 4Medical Oncology, Department of Medical and Surgical Sciences, “A. Gemelli” IRCCS, 00168 Roma, Italy; armando.orlandi@policlinicogemelli.it (A.O.); antonella.palazzo@policlinicogemelli.it (A.P.); 5Cancer Gynaecology, Department of Women, Children and Public Health Sciences, “A. Gemelli” IRCCS, 00168 Roma, Italy; ida.paris@policlinicogemelli.it; 6Plastic Surgery, Department of Women, Children and Public Health Sciences, “A. Gemelli” IRCCS, 00168 Roma, Italy; giuseppe.visconti@policlinicogemelli.it (G.V.); liliana.baroneadesi@policlinicogemelli.it (L.B.A.); marzia.salgarello@policlinicogemelli.it (M.S.)

**Keywords:** breast cancer, neoadjuvant chemotherapy, oncoplastic surgery, mastectomy, immediate breast reconstruction, aesthetic and oncological outcomes, quality of life

## Abstract

**Simple Summary:**

In patients with breast cancers larger than 2 cm undergoing neoadjuvant chemotherapy, there is a poor evaluation regarding the equivalence between oncoplastic surgery level II (OPSII) and mastectomy with immediate breast reconstruction (MIBR) regarding the aesthetic and oncological outcomes. The aim of our retrospective study was to assess whether OPSII is a safe alternative to MIBR. We confirmed the uniformity of the two techniques after neoadjuvant chemotherapy concerning loco-regional and systemic disease-free survival and overall survival in a population of 297 patients (87 undergoing OPSII, and 210 MIBR). In addition, we have highlighted how OPSII results in a lower loss of breast sensitivity after surgery and a better physical well-being of the chest. Therefore, in selected cases, OPSII should be preferred over MIBR, as it does not affect the oncological outcome, but improves physical well-being and allows the preservation of breast sensitivity in patients undergoing surgical treatment.

**Abstract:**

Oncoplastic surgery level II techniques (OPSII) are used in patients with operable breast cancer. There is no evidence regarding their safety and efficacy after neoadjuvant chemotherapy (NAC). The aim of this study was to compare the oncological and aesthetic outcomes of this technique compared with those observed in mastectomy with immediate breast reconstruction (MIBR), in post-NAC patients undergoing surgery between January 2016 and March 2021. Local disease-free survival (L-DFS), regional disease-free survival (R-DFS), distant disease-free survival (D-DFS), and overall survival (OS) were compared; the aesthetic results and quality of life (QoL) were evaluated using BREAST-Q. A total of 297 patients were included, 87 of whom underwent OPSII and 210 of whom underwent MIBR. After a median follow-up of 39.5 months, local recurrence had occurred in 3 patients in the OPSII group (3.4%), and in 13 patients in the MIBR group (6.1%) (*p* = 0.408). The three-year L-DFS rates were 95.1% for OPSII and 96.2% for MIBR (*p* = 0.286). The three-year R-DFS rates were 100% and 96.4%, respectively (*p* = 0.559). The three-year D-DFS rate were 90.7% and 89.7% (*p* = 0.849). The three-year OS rates were 95.7% and 95% (*p* = 0.394). BREAST-Q highlighted significant advantages in physical well-being for OPSII. No difference was shown for satisfaction with breasts (*p* = 0.656) or psychosocial well-being (*p* = 0.444). OPSII is safe and effective after NAC. It allows oncological and aesthetic outcomes with a high QoL, and is a safe alternative for locally advanced tumors which are partial responders to NAC.

## 1. Introduction

Oncoplastic surgery (OPS) is extending the role of breast-conserving surgery to an increasing number of patients with larger cancers who are candidates for mastectomy [1]. Oncoplastic surgery with level II techniques (OPSII) is being used with growing frequency in the multidisciplinary treatment of patients with operable breast cancer after neoadjuvant chemotherapy (NAC) [1,2,3].

OPSII includes displacement procedures, such as “inverted T mammoplasty”, “J mammoplasty”, “round block technique”, and “batwing mammoplasty”, with the reconstruction of a defect resulting from the removal of 20–50% of the native breast tissue [3,4,5,6,7,8]. The choice of surgical technique is usually based on the tumor characteristics after NAC (size and location), the extent of resection, breast features (size, shape, and glandular density), previous surgery, and the patient’s wishes and expectations [3,4,5,6,7,8,9].

Various studies report several benefits associated with the use of OPSII after NAC. This technique allows a wide excision of the tumor with safer margins, while guaranteeing appropriate cosmetic results and preventing secondary operations to correct breast deformities. It can avoid the need for mastectomy in a number of patients requiring the excision of 20–50% of the initial breast volume, without compromising local control; OPSII also allows the bypass of the higher complication rate and greater morbidity associated with mastectomy with immediate breast reconstruction (MIBR) [5,6,7,8,9,10,11,12,13].

All breast cancer patients identified as likely to require an excision between 20% and 50% of the initial breast volume after NAC should be considered for possible OPSII, as they may potentially benefit [10,11,12,13,14]. The most frequent indications for OPSII are breast cancers with a non-optimal response after NAC, for which a standard conserving surgery with safe margins would either seem impossible or lead to a major deformity, a high tumor-to-breast volume ratio, and multifocal cancer. MIBR should always be considered in patients with multicentric cancers, extensive ductal carcinoma in situ (DCIS), massive extension of microcalcification, and tumors with an unfavorable volume ratio requiring the excision of >50% of the glandular tissue.

The diffusion of OPSII in clinical practice comes from preliminary reported data that seem to indicate an adequate oncological safety and aesthetic efficacy when compared to standard breast conserving-surgery (BCS) [3,4,7,12,13,14,15,16,17,18].

However, there is still a lack of robust evidence on the role of OPSII after NAC as an alternative to MIBR for large tumors that did not respond optimally.

The aim of this study was to evaluate the safety and efficacy of OPSII after NAC by comparing oncological and aesthetic outcomes of these oncoplastic techniques to MIBR, in selected patients requiring the excision of 20–50% of the breast volume.

## 2. Materials and Methods

### 2.1. Setting and Population

This is a monocentric, retrospective study carried out on a prospectively maintained database identifying patients with locally advanced breast cancer who received NAC at the multidisciplinary breast center of the “Fondazione Policlinico Universitario Agostino Gemelli IRCCS” in Rome.

A review of the medical records collected between January 2016 and March 2021 was performed, and the breast cancer patients were divided into two cohorts based on the surgical approach: in the OPSII cohort (study group), an oncoplastic level II procedure was realized, while in the MIBR cohort (control group), a mastectomy with immediate breast reconstruction via the implant of a prothesis or expander was performed. We also included patients who reported a failure of breast reconstruction for MIBR.

Patients with a history of breast cancer or other synchronous or previous malignant neoplasms, and patients who developed metastases during NAC were not included (Figure 1).

### 2.2. Study Design

#### 2.2.1. Initial Evaluation of Patients

Patients were assessed and staged according to the TNM classification (Figure 2).

Loco-regional staging was performed by:−Clinical breast examination with the acquisition of two photographs in the frontal and lateral views of the patient’s breasts, with a mark on the skin surface depicting tumor projection and measurements [17];−Breast and axillary ultrasound (EUS);−Mammography with tomosynthesis and/or mammography with contrast medium (CESM);−Magnetic resonance image (MRI) with contrast medium;−Breast fine needle aspiration biopsy to assess the histotype and biological features; markers were positioned in the breast tissue to ensure pre-surgical localization in the case of pathological complete response (pCR) or regression to a non-palpable lesion [17];−Suspicious axillary lymph node fine needle aspiration biopsy or cytology; markers were always positioned in pathologic lymph nodes.

Systemic staging was performed by total body CT scan (TB-CT) and bone scan; positive emission tomography (PET) was used as an alternative to the two previous methods.

#### 2.2.2. Neoadjuvant Chemotherapy

The decision for neoadjuvant treatment was discussed during a multidisciplinary meeting (MDM). Patients underwent NAC according to NCCN guidelines [2].

#### 2.2.3. Operative Protocol and Surgical Technique

A complete preoperative workup including clinical assessment, ultrasonography, mammography, breast MRI, and disease staging was performed for all patients after NAC.

Surgical planning was discussed in a multidisciplinary dedicated surgery board. The main indication for OPSII was breast cancer with a non-optimal response after NAC, for which a standard conserving surgery with safe margins would either seem impossible or lead to a major deformity. MIBR was performed on patients with extensive or multicentric cancers and a tumor-to-breast volume ratio that required the excision of >50% of the glandular volume, and in the following cases: inability to obtain clear surgical margins with OPSII, contraindications to adjuvant radiotherapy, patient preference. Bilateral MIRB was performed on patients with a bilateral breast tumor or in women with unilateral disease and a high risk of contralateral breast cancer, such as BRCA mutation carriers, after tailored surgical counselling. In OPSII, when indicated, we treated the contralateral breast, performing adjustment surgery. This included reductive mastoplasty or adjustment mastopexy. In MIBR, when indicated, we performed mastopexy or breast augmentation.

A specific algorithm shared with the plastic surgeons, based on anamnestic, morphological, functional, and oncological criteria, was used to define the most appropriate surgical technique.

OPSII included “inverted T mammoplasty”, “J mammoplasty”, “round block technique”, and “batwing mammoplasty” and forecast a reconstruction of the defect resulting from the removal of between 20% and 50% of the native breast tissue. MIRB was carried out with immediate breast reconstruction via the implant of a prothesis or expander.

During surgery, we obtained a mammogram and ultrasound of the surgical specimen. We used the image to assess the distance of the residual disease or clip from the surgical margins. If the margin was close to the cancer, we conducted a re-resection of the contiguous margin to prevent the persistence of the cancer in the breast.

### 2.3. Adjuvant Treatments

The need for adjuvant treatments was discussed in an MDM and determined on the basis of patient age, pre-neoadjuvant clinical staging, surgical intervention, pathological staging, and tumor biology. Treatment protocols were performed according to NCCN guidelines [2].

#### 2.3.1. Adjuvant Chemotherapy

Patients who did not achieve a pCR to NAC were treated according to different adjuvant regimens: capecitabine was administered to patients with TN tumors, while a treatment with trastuzumab emtansine was performed in HER2+. Cancers expressing hormone receptors were treated with selective estrogen receptor modulators or luteinizing hormone release hormone analogues if of fertile age, whereas postmenopausal patients were given aromatase inhibitors.

#### 2.3.2. Adjuvant Radiotherapy

Radiotherapy was tailored based on the type of surgical intervention and staging. All patients who underwent OPSII received postoperative radiotherapy to the breast (50 Gy) with a boost (10–18 Gy) on the tumor bed. Patients who underwent MIBR were irradiated by 50 Gy to the chest wall in selected cases, according to ASCO and ASRO guidelines [19].

Axillary radiation was also considered in patients with positive lymph nodes and at a high risk of regional recurrence.

### 2.4. Follow-Up and Endpoints

Patients were evaluated every six months by surgical, oncological, or radiotherapy outpatient visits. The evaluation included clinical examination, the execution of blood chemistry tests with a panel of tumor markers (such as carcinoembryonic antigen (CEA) or CA15-3), breast EUS and mammography every six months, and systemic staging by TB-CT or PET scan every year.

The primary endpoints of our study were:−“Local disease-free survival” (L-DFS): time from the day of diagnosis to ipsilateral breast recurrence;−“Regional disease-free survival” (R-DFS): time from the day of diagnosis to ipsilateral regional lymph node recurrence;−“Distant disease-free survival” (D-DFS): time from the day of diagnosis to distant recurrence;−“Overall survival” (OS): time from the day of diagnosis to death from any cause or latest follow-up.

In addition, all patients received the BREAST-Q questionnaire (version 2.0 of the Memorial Sloan Kattering Cancer Center and the University of British Columbia) that was used to assess the aesthetic results and patient quality of life (satisfaction with breasts; psychosocial well-being; physical well-being; residual breast sensitivity). In “satisfaction with breasts”, patients express their present perception of the breast, also taking into account any post-operative complications. This survey was administered nine months after radiation therapy.

### 2.5. Statistical Analysis

All data and statistical analyses were carried out using SPSS, version 26.0. We compared physical, oncological, and treatment characteristics of the two groups in order to highlight any differences. Continuous variables were presented as means (medians and interquartile ranges (IQR), whilst categorical variables were summarized as numbers and percentages. Fisher’s exact test was used for categorical variables and an ANOVA test for continuous variables. A value of *p* < 0.05 was considered statistically significant. We then compared oncological and aesthetic outcomes between the two groups. Kaplan–Meier curves were used to plot L-DFS, R-DFS, D-DFS, and OS. The comparison between survival curves was performed using a log-rank test.

## 3. Results

Over the study period from January 2016 to March 2021, 297 breast cancer patients with locally advanced breast cancer undergoing post-NAC surgery with either OPSII or MIBR were observed. OPSII was performed in 87 cases (29.3%), while MIBR was performed in 210 cases (70.7%). Among these, 183 (87.1%) received a direct implant of a prothesis, while the remaining 27 (12.9%) received an expander initially and, subsequently, a prothesis. One patient exhibited a prosthesis infection and underwent surgical removal. Three months later, she underwent lipofilling and reconstruction with the implantation of a prosthesis. The two patient cohorts had identical characteristics in terms of histotype, subtype, grading, and T/N staging; there were statistically significant differences in age, postmenopausal status, BMI, tumor diameter, and presence of the BRCA mutation. In particular, postmenopausal status was more common among patients undergoing OPSII (49.4%) compared with the MIBR cohort (27.1%); patients undergoing OPSII showed a BMI greater than 24 in 67.1% of cases, compared to 37.7% in the MIBR group. The BRCA gene mutation was present in 49 (23.3%) mastectomy patients and 4 (4.6%) in the OPSII group; these four cases were due to patients’ choice during counseling pre-treatment, or diagnosis post-surgery. The medium tumor diameter after NAC and prior to surgery was 4.42 cm among patients undergoing OPSII, and 4.05 cm in the MIBR group (Table 1).

No differences were found in neoadjuvant treatments (Table 2). The two groups did not show statistically significant differences in clinical response; we observed a breast clinical complete response in 23 patients (26.4%) among the OPSII group, and 69 patients (32.9%) in the mastectomy group (Table 2).

No statistically significant difference was found among the two groups in terms of pathological characteristics; a pathological complete response on the breast was reported in 19 patients (21.8%) of the OPSII group and 68 patients (32.4%) undergoing mastectomy, but this difference was not statistically significant (Table 3). As regards surgical specimen margin assessment, one patient in the OPSII group and three patients in the MIBR group presented a focal micro infiltration. After multidisciplinary discussion, these patients were not referred for further surgery: the OPSII patient was treated with an adjuvant radiotherapy with a boost on the tumor bed, and the patients in the MIBR cohort were treated with adjuvant chest wall radiotherapy.

### 3.1. Adjuvant Treatment

Eighty-four patients (96.6%) subjected to OPSII received RT on the residual mammary gland, with a boost on the tumor bed. The remaining three cases refused RT. While 128 patients (61.0%) who underwent MIBR were irradiated on the chest wall, 179 patients received irradiation on III and IV lymph node levels (60.3%).

Thirty-six patients with TN residual cancer received capecitabine, while twenty-eight with HER2+ residual cancer received trastuzumab emtansine. Patients with expression of a hormone receptor received hormone therapy.

### 3.2. Oncological Outcomes

After a median follow-up of 39.5 months (range 22.8–54 months) from the diagnosis of locally advanced breast cancer, local breast recurrences (LR) had occurred in 3 (3.4%) patients who underwent OPSII and 13 (6.1%) patients treated with MIBR (Table 4).

In the MIBR cohort, LR were observed in the subcutaneous tissue near the surgical scar in nine cases, and in the axillary tail in another four patients; no LR occurred in the four patients treated with adjuvant radiotherapy for positive surgical margins. The median time to LR was 30.3 months (9.6–49.4 months). The L-DFS rates were 95.1% and 88.2% respectively in the OPSII and MIBR group. The three-year L-DFS rate was 95.1% in patients who underwent OPSII and 96.2% in mastectomy patients (*p* = 0.286) (Figure 3). The three-year cumulative risk was 0.028 for patients in the OPSII group and 0.016 for patients in the MIBR group.

Regional lymph node recurrences (RR) occurred in two (2.3%) patients subjected to OPSII and seven (3.3%) patients treated with MIBR. The median time to RR was 31.2 months (14.1–67 months). The R-DFS rates were 82.0% and 95.3% respectively in the OPSII and MIBR group. The three-year R-DFS rate was 100% in patients who underwent OPSII and 96.4% in mastectomy patients (*p* = 0.559) (Figure 4). The three-year cumulative risk was 0.000 for patients subjected to OPSII and 0.014 for patients treated with MIBR.

During follow-up, 32 patients (10.8%) developed a systemic recurrence; 10 (11.5%) distant metastases occurred in the OPSII cohort and 22 (10.5%) in the MIBR group. The D-DFS rates were 81.8%% and 84.3%% in the OPSII and MIBR groups, respectively. The three-year D-DFS rate was 90.7% among patients who underwent OPSII and 89.7% in the MIBR cohort (*p* = 0.849) (Figure 5).

We observed the death of 19 (6.4%) women, 4 cases in the OPSII group and 15 in MIBR cohort; death was attributed to breast cancer in all cases. The OS rate was 93.4% and 87.9% respectively in the OPSII and MIBR group. The three-year OS rate was 95.7% among patients who underwent OPSII and 95% in the MIBR cohort (*p* = 0.394) (Figure 5).

In a univariate and multivariate analysis concerning D-DFS (Table 5), HER2+ with pCR on breast, pCR on axilla, and RT were significantly associated with distant relapse. However, only HER2+ remained significantly associated with relapse at multivariate analysis, with a protective role.

### 3.3. Aesthetic Outcomes and Health-Related Quality of Life

A total of 194/297 (65.3%) patients completed our survey assessing their postoperative aesthetic results and quality of life (28.4% and 71.6%, respectively, for the OPSII and MIBR groups) (Table 6).

A statistically non-significant difference was shown for satisfaction with breasts. The average score was 61 for OPSII (Figure 6) and 51.6 for MIBR (Figure 7)—*p* = 0.656. Additionally, in psychosocial well-being (average score 64.2 vs. 58.1—*p* = 0.444), there was no significant difference among the two groups.

Statistically significant (*p* < 0.05) advantages in terms of physical well-being with minor chest pain and preserved breast skin sensitivity were observed in the OPSII group.

No differences emerged in the spheres of ordinary and sexual life between the OPSII and MIBR groups (*p* = 0.621 and *p* = 0.499, respectively).

## 4. Discussion

NAC is being used with increasing frequency in the multidisciplinary treatment of patients with locally advanced breast cancer. Factors favoring NAC include high tumor-to-breast volume, lymph node-positive disease, and specific biological features of primary cancer (high grade, hormone receptor-negative, HER2+, TN cancer) [2,18,19,20,21,22,23].

A key benefit of NAC is the downstaging of tumors, which favors breast-conservative surgery (BCS) over MIBR and reduces surgical morbidity [2,18].

The two main goals of the surgeon when performing BCS after NAC are to obtain tumor-free margins and an adequate aesthetic outcome; tumor-involved margins must be avoided because this condition increases the risk of local-regional recurrence and therefore requires additional local therapy, such as a radiation therapy boost, re-excision, or even mastectomy.

In order to optimize the oncological and aesthetic outcomes in selected patients with large or multifocal tumors desiring breast conservation, OPSII can be used after NAC. These techniques allow the resection of a greater amount of breast tissue with safer margins and appropriate cosmetic results, avoiding the need for mastectomy in patients with a partial response.

The diffusion of these techniques in clinical practice after NAC is due to the preliminary results of some retrospective and prospective studies obtained by comparing OPSII to standard BCS.

There are currently limited studies that have verified the oncological safety and aesthetic efficacy of OPSII compared to MIBR for large tumors that did not respond optimally after NAC [20,21,22,23,24,25,26,27].

In a recent cohort study, Van La Parra and colleagues analyzed a consecutive series of 65 patients who underwent OPS (study group) after NAC for large breast cancer compared with 130 matched patients treated by NAC, followed by standard BCS in 65 cases and mastectomy in 65 cases (two case-controlled groups) [28]. The authors concluded that OPS is safe after NAC for large breast cancers, and provides excellent local control, identical to that of tumors with a better response treated by standard BCS; after a mean follow-up of 59 months, the five-year local recurrence rates were 0%, 0%, and 10.5% while the five-year overall survival was 85.3%, 94.1%, and 79.9% respectively for the OPS, standard BCS, and mastectomy cohorts [28].

Mehmet A. Gulcelik and Lutfi Dogan also reported no significant differences between patient groups who underwent planned or unplanned OPS, or mastectomy after NAC, in terms of long-term local recurrence-free survival, disease-free survival, and overall survival [29].

In order to evaluate the feasibility of OPSII for large tumors that require the excision of 20–50% of the breast volume after NAC completion, we compared 87 patients treated with oncoplastic techniques versus 210 patients treated with MIBR.

Our study shows that OPSII is a safe and effective alternative to MIBR, as it allows the obtainment of adequate oncological and aesthetic outcomes and a good patient quality of life.

As regards oncological safety, the results between OPSII and MIBR, after a median follow-up of 39.5, were similar and there were no statistically significant differences in terms of disease-free survival and overall survival; the LR rates were 3.4% and 6.1% respectively for the OPSII and MIBR group. The L-DFS rates were 95.1% and 88.2% in the OPSII and MIBR groups, respectively. The three-year L-DFS rate was 95.1% in patients who underwent OPSII and 96.2% in the mastectomy group (*p* = 0.286).

As regards aesthetic outcomes and patient quality of life, the assessment of the BREAST-Q questionnaire in our study showed similar results between patients treated with OPSII and MIBR; a statistically significant difference was observed only in physical well-being with more chest pain and minor breast sensitivity in the MIBR group compared with patients treated with OPSII. A statistically non-significant difference in favor of the OPSII group was also observed for satisfaction with breasts and psychosocial well-being.

However, we think that some recommendations are mandatory for the success of OPSII in large tumors that did not respond optimally to NAC [28].

A multidisciplinary discussion, in a dedicated “surgery meeting” with a careful patient assessment and disease staging, is essential to select the best candidates for OPSII; OPSII should never be performed in patients with multicentric cancers, extensive ductal carcinoma or a tumor-to-breast volume ratio that requires the excision of >50% of the glandular volume, or in small breasts with minor ptosis and previous radiotherapy. Other clinical factors can contribute to the decision process; in our study, age, postmenopausal status, BMI, tumor diameter, and presence of BRCA mutation significantly differed between the OPSII and MIBR cohort, and were probably factors in determining the choice of surgery.

The use of breast tattooing and the placement of clips before chemotherapy should always be performed to mark the primary tumor site and define its extension, in order to help the surgeon in the subsequent surgical planning. The choice of the more appropriate oncoplastic technique should be tailored to each patient and based on the tumor characteristics (size and location), the extent of resection, and the breast characteristics (size, shape, and glandular density). Specific algorithms can assist the breast surgeon in the decision process. An accurate intraoperative radiological and pathological evaluation of the specimen is essential in order to obtain tumor-free margins while keeping the amount of healthy breast tissue excised as low as possible; systematic circumferential tumor cavity shaving to have a backup to the lumpectomy margins, and the placement of clips within the excision cavity as a “landmark” to define the tumor bed and guide adjuvant breast radiotherapy, should always be performed. The correct performance of adjuvant radiotherapy according to the guidelines is mandatory to complete the conservative treatment and to minimize the risk of local recurrence [30].

To date, our study constitutes one of the largest comparative patient series in the literature between OPSII and MIBR after NAC; however, it presents some limitations. It is a retrospective analysis of a single institution. Thus, we have a limited number of patients treated with OPSII compared to MIBR. Furthermore, it has a short follow-up duration. Therefore, additional high-quality multicenter studies are required to compare short- and long-term outcomes and overcome the aforementioned limitations.

## 5. Conclusions

Our experience shows that OPSII is safe and effective after NAC, as it allows the achievement of adequate oncological and aesthetic outcomes with a high patient quality of life, similarly to MIBR; OPSII can be considered a valuable surgical alternative to MIBR for selected locally advanced tumors that do not show an optimal response after primary chemotherapy, and that require the excision of 20–50% of the glandular volume.

## Figures and Tables

**Figure 1 cancers-14-01275-f001:**
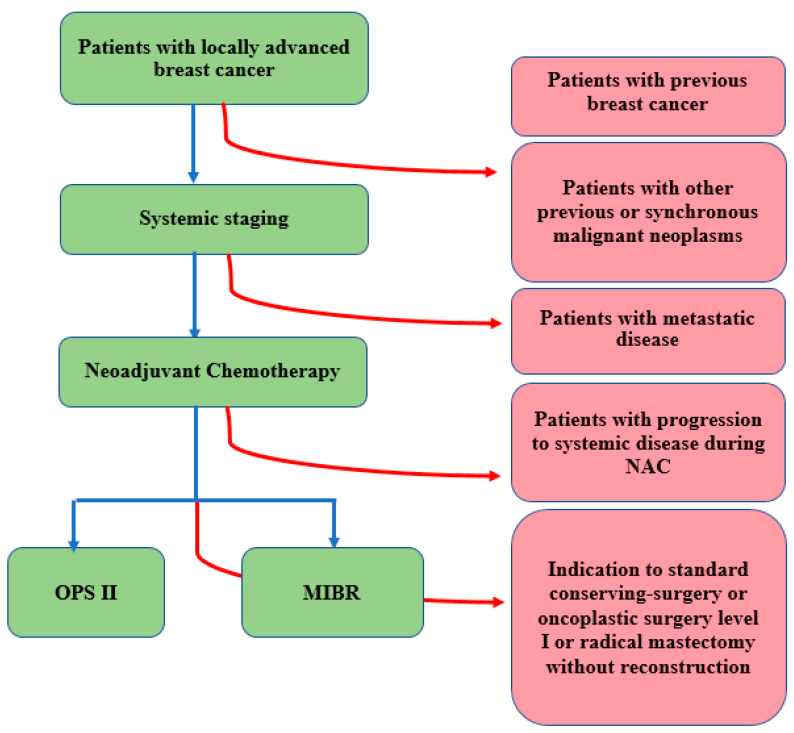
Inclusion and exclusion flow diagram.

**Figure 2 cancers-14-01275-f002:**
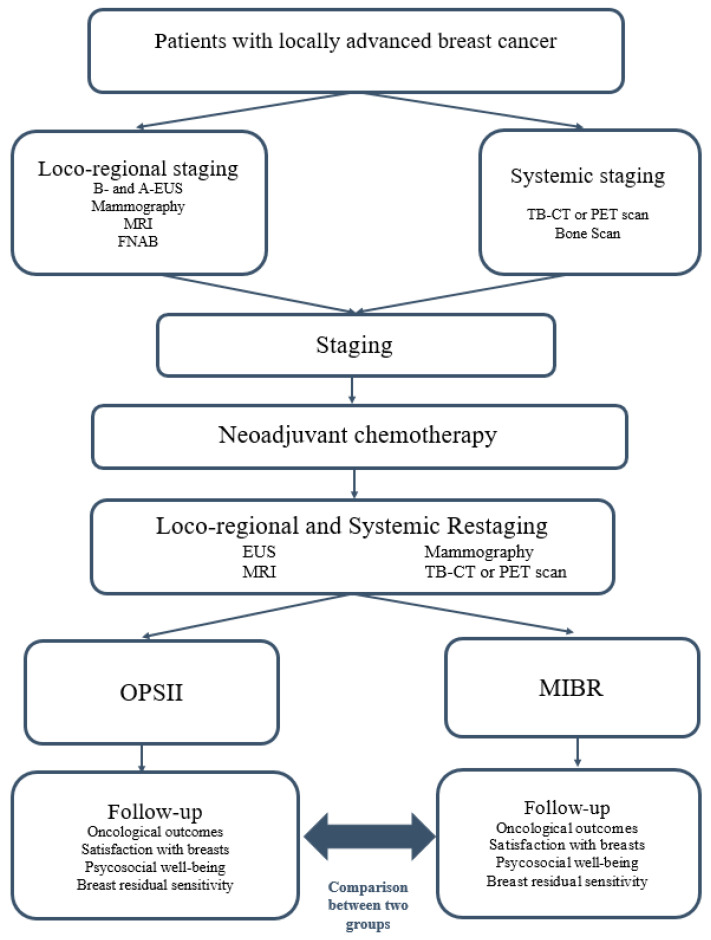
Diagnostic and therapeutic flow chart. (B- and A-EUS: breast and axillary ultrasound).

**Figure 3 cancers-14-01275-f003:**
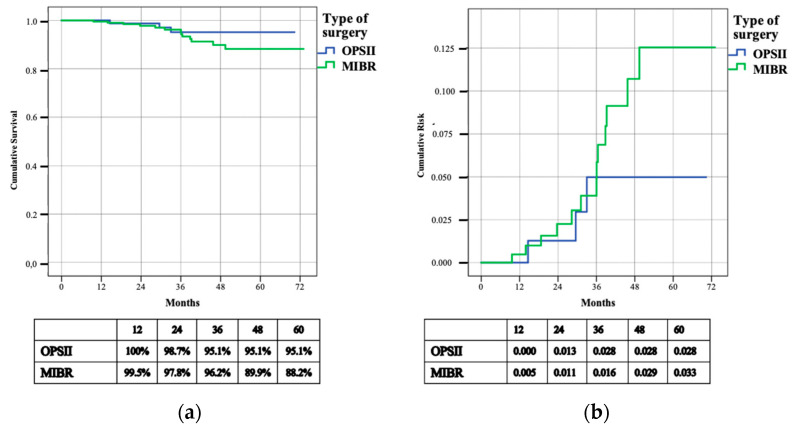
Cumulative survival (**a**) and risk (**b**) of local recurrence (L-DFS).

**Figure 4 cancers-14-01275-f004:**
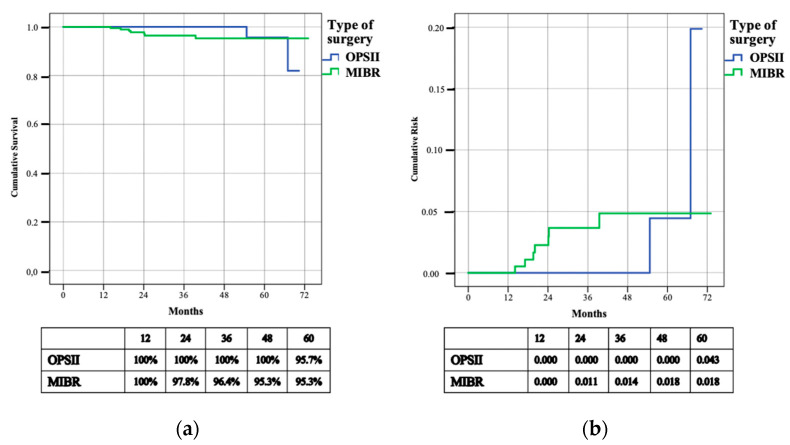
Cumulative survival (**a**) and risk (**b**) of regional recurrence (R-DFS).

**Figure 5 cancers-14-01275-f005:**
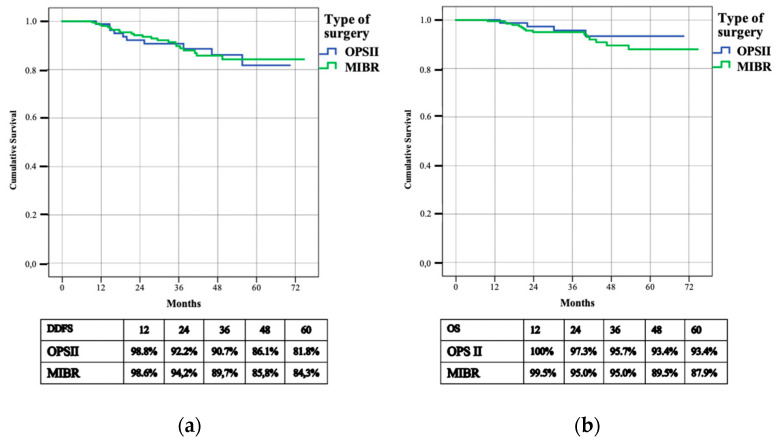
Distant disease free survival (D-DFS) (**a**) and overall survival (OS) (**b**).

**Figure 6 cancers-14-01275-f006:**
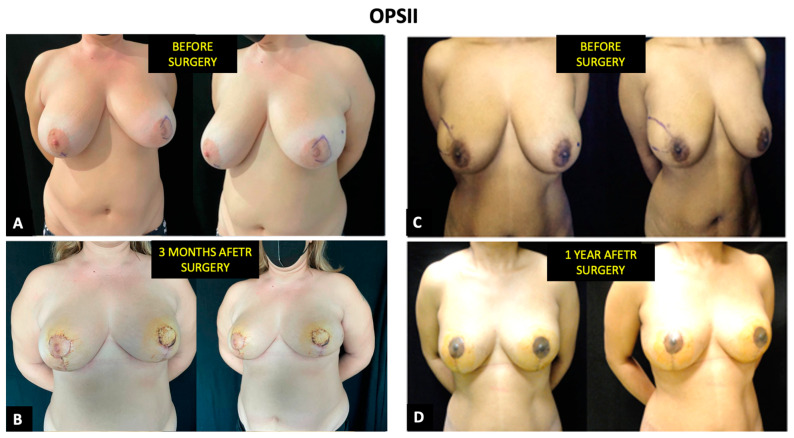
Evaluation of aesthetic outcomes in two patients who underwent OPSII. The view is given before surgery (**A**) and (**C**), after 3 months (**B**), and after one year (**D**).

**Figure 7 cancers-14-01275-f007:**
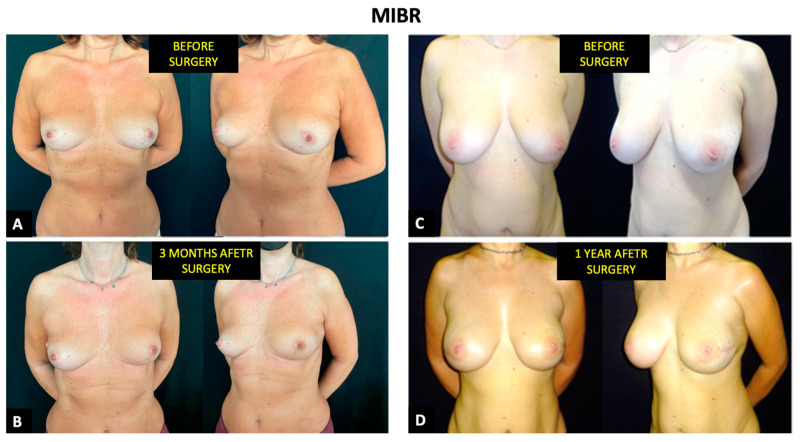
Evaluation of aesthetic outcomes in two patients who underwent MIBR. The view is given before surgery (**A**) and (**C**), after 3 months (**B**), and after one year (**D**).

**Table 1 cancers-14-01275-t001:** Clinical characteristic of patients before NAC.

Characteristics	All Patients	OPSII	MIBR	*p*-Value
	297	87 (29.3%)	210 (70.7%)	
Age (y)	46.3 (44.8; 39.7–52)	50.1 (48.5; 42.7–55.8)	44.6 (43.5; 38.6–50.5)	0.000
Postmenopausal status - Yes - No	100 (33.3%) 197 (65.7%)	43 (49.4%) 44 (50.6%)	57 (27.1%) 153 (72.9%)	0.000
BMI (Kg/m^2^)	23.7 (23.4; 21.2–25.1)	27.1 (26.6; 23.9–30.1)	23.7 (23.4; 21.2–25.1)	0.006
BMI classes - BMI < 18 - BMI 18–24 - BMI 24–30 - BMI > 30	3 (1.3%) 155 (52.1%) 103 (34.6%) 36 (12%)	0 (0%) 27 (29.9%) 38 (44.8%) 22 (25.3%)	3 (1.4%) 128 (60.9%) 65 (31%) 14 (6.7%)	0.000
Breast related cancer antigens (BRCA) mutations	53 (17.8%)	4 (4.6%)	49 (23.3%)	0.000
Histotype - Ductal invasive carcinoma - Lobular invasive carcinoma - Other	198 (66.7%) 22 (7.4%) 77 (25.9%)	61 (70.1%) 6 (6.9%) 20 (23%)	137 (65.2%) 16 (7.6%) 57 (27.1%)	0.715
Tumor subtype - Luminal A - Luminal B - HER2+ - TN	14 (4.7%) 133 (44.8%) 98 (33%) 52 (17.5%)	8 (9.2%) 40 (46%) 24 (27.6%) 15 (17.2%)	6 (2.9%) 93 (44.3%) 74 (35.2%) 37 (17.6%)	0.095
Grading - G1 - G2 - G3	4 (1.3%) 94 (31.7%) 199 (67%)	3 (3.4%) 26 (29.9%) 58 (66.7%)	1 (0.5%) 68 (32.4%) 141 (67.1%)	0.125
Tumor diameter (mm)	41.6 (35; 27–50)	44.2 (38; 27–56)	40.5 (35; 26.7–50)	0.038
Clinical T - cT2 - cT3 - cT4	215 (72.4%) 64 (21.5%) 18 (6.1%)	59 (67.8%) 20 (23%) 8 (9.2%)	156 (74.3%) 44 (21%) 10 (4.8%)	0.291
Multifocality/multicentricity	159 (53.5%)	49 (56.3%)	110 (52.4%)	0.609
Clinical N - cN0 - cN1 - cN2 - cN3	111 (37.4%) 131 (44.1%) 44 (14.8%) 11 (3.7%)	29 (33.3%) 43 (49.4%) 14 (16.2%) 1 (1.1%)	82 (39%) 88 (41.9%) 30 (14.3%) 10 (4.8%)	0.304

**Table 2 cancers-14-01275-t002:** Schemes of delivered neoadjuvant treatments and related clinical response.

Characteristics	All Patients	OPSII	MIBR	*p*-Value
	297	87 (29.3%)	210 (70.7%)	
Neoadjuvant chemotherapy - Anthracycline and/or taxanes - Anthracycline + taxanes - Other schemes	5 (1.7%) 235 (79.1%) 57 (19.2%)	0 (0%) 73 (83.9%) 14 (16.1%)	5 (2.3%) 162 (77.2%) 43 (20.5%)	0.385
Regimens with trastuzumab	98 (33%)	24 (27.6%)	74 (35.2%)	0.096
Clinical response on T - Clinical complete response - Clinical partial response - No response - Progression	92 (30.9%) 189 (63.7%) 8 (2.7%) 8 (2.7%)	23 (26.4%) 61 (70.1%) 2 (2.3%) 1 (1.1%)	69 (32.9%) 128 (61%) 6 (2.9%) 7 (3.3%)	0.425
ycT - 0 - 1 - 2 - 3	92 (31%) 111 (37.4%) 79 (26.5%) 15 (5.1%)	23 (26.4%) 37 (42.5%) 25 (28.7%) 2 (2.3%)	69 (32.9%) 74 (35.2%) 54 (25.7%) 13 (6.2%)	0.290
Clinical Response on N - ycN0 - ycN+	231 (77.8%) 66 (22.2%)	66 (75.9%) 21 (24.1%)	165 (78.6%) 45 (21.4%)	0.646

**Table 3 cancers-14-01275-t003:** Pathological characteristic of all patients, according to the type of surgery.

Characteristics	All Patients	OPSII	MIBR	*p*-Value
	297	87 (29.3%)	210 (70.7%)	
ypT - ypT0 - ypT0i+ ^1^ - ypTmic ^2^ - ypT1 - ypT2 - ypT3 - ypT4	87 (29.3%) 6 (2%) 24 (8%) 113 (38.1%) 56 (18.9%) 9 (3%) 2 (0.7%)	19 (21.8%) 0 (0%) 4 (4.6%) 39 (44.9%) 20 (23%) 4 (4.6%) 1 (1.1%)	68 (32.4%) 6 (2.9%) 20 (9.5%) 74 (35.2%) 36 (17.1%) 5 (2.4%) 1 (0.5%)	0.074
Mean residual tumor size (mm)	11.2 (6; 0–18)	14.2 (10; 1–23)	10 (4.5; 0–15.3)	0.168
Pathological response on T - Pathological complete response - Pathological partial response - Pathological progression or no response	87 (29.3%) 185 (62.3%) 25 (8.4%)	19 (21.8%) 57 (65.5%) 11 (12.7%)	68 (32.3%) 128 (61%) 14 (6.7%)	0.074
Multifocality/multicentricity	91 (30.3%)	29 (33.3%)	62 (29.5%)	0.383
Histotype in residual disease - Ductal invasive carcinoma - Lobular invasive carcinoma - Others - Not evaluable ^3^	159 (53.5%) 18 (6.1%) 31 (10.4%) 89 (30%)	53 (60.9%) 6 (6.9%) 9 (10.4%) 19 (21.8%)	106 (50.3%) 12 (5.7%) 22 (10.5%) 70 (33.3%)	0.233
ER - Positive - Negative - Not evaluable ^3^	138 (46.5%) 49 (16.5%) 110 (37%)	44 (50.6%) 17 (19.5%) 26 (29.9%)	97 (46.2%) 31 (14.8%) 82 (39%)	0.281
PR - Positive - Negative - Not evaluable ^3^	97 (32.7%) 90 (30.3%) 110 (37%)	31 (35.6%) 30 (34.5%) 26 (29.9%)	68 (32.4%) 60 (28.6%) 82 (39%)	0.313
Ki67 - ≥25 - <24 - Not evaluable ^3^	80 (26.9%) 107 (36.1%) 110 (37%)	25 (28.7%) 36 (41.4%) 26 (29.9%)	55 (26.2%) 73 (34.8%) 82 (39%)	0.227
Tumor subtype - Luminal A - Luminal B - HER2+ - TN - Not evaluable ^3^	92 (31%) 38 (12.8%) 28 (9.4%) 36 (12.1%) 103 (34.7%)	29 (33.3%) 14 (16.1%) 5 (5.7%) 16 (18.5%) 23 (26.4%)	61 (29.1%) 24 (11.4%) 23 (11%) 20 (9.5%) 82 (39%)	0.052
N patients who underwent SLNB Mean number of lymph nodes removed (range)	217 (73.1%) 2.86 (3; 2–4)	64 (73.6%) 2.8 (3; 1.25–4)	153 (72.9%) 2.88 (3; 2–4)	0.492
ypN (sn) - ypN0 - presence of ITC - presence of mic - 1 lymph node positive - 2 lymph nodes positive - 3 lymph nodes positive	131 (60.4%) 14 (6.5%) 16 (7.4%) 42 (19.4%) 7 (3.2%) 7 (3.2%%)	32 (50%) 3 (4.7%) 5 (7.8%) 16 (25%) 3 (4.7%) 5 (7.8%)	99 (64.7%) 11 (7.2%) 11 (7.2%) 26 (17%) 4 (2.6%) 2 (1.3%)	0.073
N patients who underwent AD Mean number of lymph node removed (range)	194 (65.3%) 13.8 (13; 10–16.25)	61 (70.1%) 14.6 (13; 10–17)	133 (63.3%) 13.4 (12; 10–16)	0.289
ypN - ypN0 - ypN0i+ - ypNmic - ypN1 - ypN2 - ypN3	59 (30.4%) 10 (5.1%) 12 (6.2%) 67 (34.6%) 24 (12.4%) 22 (11.3%)	17 (27.8%) 2 (3.2%) 2 (3.2%) 23 (37.8%) 10 (16.5%) 7 (11.5%)	42 (31.6%) 8 (6%) 10 (7.5%) 44 (33.1%) 14 (10.5%) 15 (11.3%)	0.388

^1^ Evidence of isolated cancer cells in the lymph node. ^2^ Evidence of microscopic residual of tumor (<0.2 mm) in the lymph node. ^3^ No residual disease, ITC, or mic.

**Table 4 cancers-14-01275-t004:** Oncological outcomes.

Characteristics	All Patients	OPSII	MIBR	Long-Rank
	297	87 (29.3%)	210 (70.7%)	
Surgical margins - Clear margins - “ink on tumor”	293 (98.7%) 4 (1.3%)	86 (98.9%) 1 (1.1%)	207 (98.6%) 3 (1.4%)	0.949
Local disease free survival - N. patients with breast recurrence - 3 years—L-DFS - 3 years cumulative risk	16 (5.4%) 95.8% 0.014	3 (3.4%) 95.1% 0.028	13 (6.1%) 96.2% 0.016	0.286
Regional disease free survival - N. patients with axillary recurrence - 3 years—R-DFS - 3 years cumulative risk	9 (3%) 97.5% 0.010	2 (2.3%) 100% 0.000	7 (3.3%) 96.4% 0.014	0.559
Distant disease free survival - N. patients with systemic recurrence - 3 years—D-DFS	32 (10.8%) 90%	10 (11.5%) 90.7%	22 (10.5%) 89.7%	0.849
Overall survival - N. patients deceased - 3 years—OS	19 (6.4%) 95.2%	4 (4.6%) 95.7%	15 (7.1%) 95%	0.394

**Table 5 cancers-14-01275-t005:** Univariate and multivariate analysis for distant disease free survival.

Characteristics	Univariate Analysis			Multivariate Analysis			
	*p*-Value	OR	95% CI	*p*-Value	OR	95% CI	B-Coefficient
Postmenopausal status	0.607	1.221	0.571–2.611				
BRCA pathological mutations	0.271	1.622	0.685–3.841				
Grading 3 (G3)	0.813	1.100	0.499–2.424				
cT (3 or 4)	0.193	1.663	0.773–3.557				
cN+	0.064	2.285	0.954–5.475				
HER2+	0.005	0.056	0.007–0.413	0.015	0.082	0.011–0.617	−2.506
TN	0.089	2.068	0.895–4.782				
pCR on breast	0.009	0.068	0.009–0.506	0.054	0.135	0.018–1.037	−2.004
pCR on axilla	0.007	1.685	1.156–2.455	0.163	1.308	0.897–1.909	0.269
Ink on tumor	0.999	0.000	0.000				
ypT (3 or 4)	0.089	3.310	0.832–13.176				
ypN (2 or 3)	0.574	1.249	0.575–2.714				
Radiotherapy ^1^	0.009	2.705	1.289–5.675	0.133	1.568	0.872–2.820	0.450

^1^ radiotherapy on chest wall in MIBR performed on 128 patients (61.0%) only.

**Table 6 cancers-14-01275-t006:** Aesthetic outcomes and loss of sensitivity according to the type of surgery.

Characteristisc	All Patients	OPSII	MIBR	*p*-Value
	297	87 (29.3%)	210 (70.7%)	
Number of answers	194 (65.3%)	55 (28.4%)	139 (71.6%)	
Q.1 Satisfaction with breasts				
Average score (median score) ^1^	54.3 (53)	61 (58)	51.6 (53)	0.656
Score				
- 0–34	36 (18.6%)	5 (9%)	31 (22.3%)	
- 39–58	88 (45.4%)	25 (45.5%)	63 (45.3%)	0.052
- 63–100	70 (36.1%)	25 (45.5%)	45 (32.4%)	
Q.2 Psychosocial well-being				
Average score (median score) ^1^	59.8 (56)	64.2 (64)	58.1 (55)	0.444
Score				
- 0–39	28 (14.4%)	4 (7.3%)	24 (17.3%)	
- 41–58	73 (37.6%)	19 (34.5%)	54 (38.8%)	0.105
- 60–100	93 (47.9%)	32 (58.2%)	61 (43.9%)	
Q.3 Physical well-being: chest				
Average score (median score) ^1^	37 (32)	28.6 (28)	40.3 (3)	0.007
Score				
- 0–32	103 (53.1%)	35 (63.6%)	68 (48.9%)	
- 36–68	66 (34%)	18 (32.7%)	48 (34.5%)	0.027
- 72–100	25 (12.9%)	2 (3.6%)	23 (16.5%)	
Q.4 Loss of sensitivity				
- No	46 (23.7%)	28 (50.9%)	18 (12.9%)	0.000
- Yes	148 (76.3%)	27 (49.1%)	121 (87.1%)	
Q.4.1 Percentage of sensitivity loss				
- Mean percentage loss	7.49 (8)	6.44 (7)	7.73 (8)	0.631
Score				
- 10–30	7 (4.7%)	4 (14.8%)	3 (2.5%)	
- 40–70	53 (35.8%)	12 (44.4%)	41 (33.9%)	0.011
- 80–100	88 (59.5%)	11 (40.7%)	77 (63.6%)	
Q.4.2 Influence of sensitivity loss on ordinary life				
- Mean influence	4.15 (5)	3.56 (3)	4.28 (5)	0.784
Score				
- 0–30	62 (41.9%)	14 (51.9%)	48 (39.7%)	
- 40–70	60 (40.5%)	9 (33.3%)	51 (42.1%)	0.621
- 80–100	26 (17.6%)	4 (40.7%)	22 (18.2%)	
Q.4.3 Influence of sensitivity loss on sex life				
- Mean influence	6.06 (7)	6.11 (7)	6.05 (7)	0.260
Score				
- 0–30	37 (25%)	5 (18.5%)	32 (26.4%)	
- 40–70	39 (26.4%)	10 (37.1%)	29 (24%)	0.499
- 80–100	72 (48.6%)	12 (44.4%)	60 (49.6%)	

^1^ according to BREAST-Q version 2.0.

## Data Availability

The data that support the findings of this study are available on request from the corresponding author after anonymizing the identifiable information.

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
