# Peer review of "Level II Oncoplastic Surgery as an Alternative Option to Mastectomy with Immediate Breast Reconstruction in the Neoadjuvant Setting: A Multidisciplinary Single Center Experience"

_cancers, 2022, doi:10.3390/cancers14051275_

Round 1

Reviewer 1 Report

I read the manuscript with interest as breast cancer is a fundamental part of my practice. The authors show their experience on conservative surgery with patient oncoplastic techniques undergoing neoadjuvant chemotherapy for breast cancer.

My main comments are as follows:

  1. The goals are broad and radical
  2. Breast cancer overview and methods are to be improved and relevant to the purpose of the research, I suggest to keep the methods targeted so that readers can stay focused on the research question, these could be shortened and help reduce the paper. the section on chemotherapy would omit this detailed description.
  3. Please clarify what has been done in patients with positive margins.
  4. Explain the management of the non-diseased breast
  5. Reference 19? Check
  6. The figures and tables are easy to interpret, these could be shortened too much non-useful information, add some figures on aesthetic outcomes

Author Response

Thanks for the review and for the suggestions. We have implemented the text following your suggestions.

Reviewer 2 Report

Dear authors,

General commentary. In my opinion the article is too long, there are some parts with too many details (for example, Materials and methods, points 2.22, it could be shortened because I suppose this is according to well-known guidelines. My recommendation is to reduce the article extension.

Simple summary. You mention “conservative mastectomy” after that in the article you use mastectomy; I think that you should change that and use the same term “Mastectomy” in the sinople summary.

Introduction. You do not mention “replacement volume” in breast conservative oncoplastic techniques.

Oncoplastic techniques extend breast-conserving surgery to patients with neoadjuvant chemotherapy response unfit for conventional techniques.

Regaño S, Hernanz F, Ortega E, Redondo-Figuero C, Gómez-Fleitas M.

World J Surg. 2009 Oct;33(10):2082-6. doi: 10.1007/s00268-009-0152-x.

It was one of the first published articles about the topic you are managing. ¿Do you not consider these techniques as a Level II?

Material and methods.

About patient inclusion and exclusion, all patient who undergone MIBR were included, were there some fail reconstructions because implant infection, extrusion, etc.?

In all text you mention the percentage of breast volume excision but there is not comment that hod did you do this estimation, even though there is not any data about breast volume.

Which version and module of BREAST-Q questionnaire were used? Did you use the same for both cohorts? This point should be clarified.

Results.

Table 4. “on ink “tumor can be a mistake, “no ink” I suppose

I congratulate you for the really very low rate of affected margins, even though if it considered that you are treated patients after neoadjuvant chemotherapy.

You should mention in the material how do you manage the test or control of surgical margins.

Point 3.2

You should give the time when the survey was done and better from the end of radiotherapy in the breast conservative cohort.

Conclusions.

“the excision of 20-50%.....

It should be removed, it is cannot be a conclusion of your work

You should add a part aboul limitations of your work

Retrospective one, possible bias, estimation of size sample to observe differences ( your event are very low), etc.

Author Response

(The authors gave the same response as above.)

Reviewer 3 Report

I appreciate the paper. Some serious flaws need to be eliminated:

  1. too many authors. I doubt that 21 authors are required for this single center study. please reduce to the necessary amount
  2. the section on neoadjuvant chemo is not clear. why is carboplatin added in her2/neu positive cancers? why AC? this is in contrast to TRAIN-2 study. I would omit this detailed description. it does not add much information. 
  3. please clarify menopausal status into pre or post
  4. please offer an interpretation why postop the her2 status in the groups is different
  5. please show the number of patients in the mastectomy group who have received radiotherapy. a multivariate analysis should show if this factor is related to any kind of recurrence. currently, there is only a short section in the methods. please indicate why the patients who had positive margins where not surgically treated (ok in micro focal R1, but needs explanation)
  6. it is totally inappropriate to just cite "19" on the mastectomy surgery as "19" is a radiotherapy study. the type of reconstruction should be mentioned, implant/expander/autologous?
  7. oncoplastic surgery and reconstruction always include the management of the contralateral side. please include how this was managed besides the BRCA patients. Was the contralateral side adjusted? 
  8. why tumor markers? in addition they are not mentioned anywhere anymore
  9. Discussion: citation of "Mehmet A. Gulcelik and Lutfi Dogan" sounds awkward and is not appropriate (it is not "29"!). please check all citations 

Author Response

(The authors gave the same response as above.)

Round 2

Reviewer 2 Report

Dear Editor, I read the new version of the article , in this form it can be accepted for publication

Reviewer 3 Report

The authors have followed most of my suggestions and have the improved the paper. There are some details which were not changed but all explained. The question on tumor markers was related to laboratory work. So there was a misunderstanding. 

Overall I think the paper should be accepted without further revisions.